# A Novel Co-Crystal of Bexarotene and Ligustrazine Improves Pharmacokinetics and Tissue Distribution of Bexarotene in SD Rats

**DOI:** 10.3390/pharmaceutics12100906

**Published:** 2020-09-23

**Authors:** Shuyue Ren, Lingtai Jiao, Shiying Yang, Li Zhang, Junke Song, Haoying Yu, Jingrong Wang, Tingting Lv, Lan Sun, Yang Lu, Guanhua Du

**Affiliations:** 1Institute of Materia Medica, Chinese Academy of Medical Sciences, Peking Union Medical College, 1 Xian Nong Tan Street, Beijing 100050, China; renshuyue@imm.ac.cn (S.R.); zhangbx@imm.ac.cn (L.J.); ysy@imm.ac.cn (S.Y.); zhangl@imm.ac.cn (L.Z.); smilejunke@imm.ac.cn (J.S.); yhy@imm.ac.cn (H.Y.); wangjingrong@imm.ac.cn (J.W.); lvtingting@imm.ac.cn (T.L.); 2Beijing Key Laboratory of Drug Targets Identification and Drug Screening, 1 Xian Nong Tan Street, Beijing 100050, China; 3Beijing Key Laboratory of Polymorphic Drugs, Center of Pharmaceutical Polymorphs, 1 Xian Nong Tan Street, Beijing 100050, China

**Keywords:** pharmaceutical co-crystals, active pharmaceutical ingredients, bexarotene, solubility, bioavailability, cerebral distribution

## Abstract

Bexarotene (BEX), a specific retinoic acid X receptor (RXR) agonist granted by Food and Drug Administration (FDA) approval for the clinical treatment of T cell lymphoma, has now been found to exert pharmacological effects in the nervous system, with low bioavailability and poor cerebral distribution limiting its application in treatment on neurological disorders. Pharmaceutical co-crystal was a helpful method to improve the bioavailability and tissue distribution of active pharmaceutical ingredients (APIs). Here, 2bexarotene-ligustrazine (2BEX-LIG), a novel co-crystal system of BEX and ligustrazine (LIG) of which with BEX is an API, was constructed with satisfactory stability and enhanced solubility. The pharmacokinetics characteristics of BEX were detected, and the results showed that the absolute bioavailability and the cerebral concentration of BEX in rats administrated with 2BEX-LIG were enhanced from 22.89% to 42.86% and increased by 3.4-fold, respectively, compared with those in rats administrated an equivalent of BEX. Hence, our present study indicated that the novel co-crystal of 2BEX-LIG contributed to improving BEX oral bioavailability and cerebral distribution, thereby providing significant advantages for clinical application of brain tumors and other neurological diseases.

## 1. Introduction

Co-crystals have attracted increasing interest in a variety of fields because of their superior properties as multicomponent materials that are rationally designed through the crystal engineering principle [1,2,3]. Pharmaceutical co-crystals are potentially attractive to improve physicochemical properties of active pharmaceutical ingredients (APIs), including stability [4,5,6], solubility [7,8,9,10], bioavailability [11,12,13], and tissue distribution. It is difficult to form salts for APIs without ionizable moieties, whereas co-crystals can be used as an alternative method to generate novel solid forms to improve the physicochemical properties. Moreover, when compared with salt, more APIs are available for co-crystal, which increases the possibility for novel combinations with synergistic effects [14].

Bexarotene (BEX) is a novel oral-specific agonist of retinoid X receptors (RXRs) approved by the United States Food and Drug Administration in 1999 for cutaneous T-cell lymphomas (CTCL). In addition, BEX has been used for the treatment of non-small cell lung cancer (NSCLC) [15], triple negative breast cancer (TNBC) [16], glioblastoma multiforme (GBM) [17], and thyroid cancer [18]. Recently, BEX has been found to exert pharmacological effects on nervous system disorders such as poststroke [19], brain injury [20], and neurodegenerative diseases [21]. However, BEX is an API with a poor solubility of 0.0003 g/L [22,23] in pure water. The poor solubility of BEX led to low bioavailability in rats, which was about 18.13% in a previous study [24]. Although BEX was formulated into linoleic acid or sunflower oil [25] to improve the physicochemical properties, the oral bioavailability of BEX in rats was still quite low, at 31.5% and 31.4%, respectively. However, due to the poor water solubility and bioavailability, the clinical practice of BEX is severely restricted [26]. In particular, the low distribution of BEX in the cerebrum leads to a relatively high dose of BEX which was needed in previous studies treating patients with degenerative diseases [27]. Besides, the United States Food and Drug Administration-approved BEX for treating CTCL caused side effects and may present tolerability problems in patients [28,29]. Previous studies reported two different BEX nanocrystals which improved the area under the curve (AUC _(0–∞)_) of BEX about by 2-fold [24] and 3-fold [30,31], respectively, but required optimal stabilizers and appropriate reaction temperature, separately. Besides, cerebral concentration of BEX was not detected in the above studies. Therefore, there is a continued need to investigate suitable formation to improve the bioavailability and cerebral distribution of BEX.

Ligustrazine (LIG, also known as tetramethylpyrazine, TMP), the major pharmacological ingredient of Rhizoma Chuanxiong [32], is a kind of alkaloid [33]. Previous studies have revealed that LIG had various pharmacological activities, which has been widely used in the treatment of cerebral ischemic injury and conferring neuroprotection [34]. LIG could promote postischemic neuroregeneration, can recover neurological function in rats [35], and had a neuroprotective function via Bax/Bcl-2 and the caspase-3 pathway in PC12 cells and rats with vascular dementia [36]. In addition, LIG was reported to form cocrystal with a series of carboxylic acids, which were measured by the complete 14N Nuclear Quadrupole Resonance (NQR) spectra [37,38] because of its outstanding pharmacokinetic characteristics, such as rapid absorption into the blood, crossing of the blood–brain and blood–labyrinth barriers [39], broad distribution, and no accumulated toxic effect [40].

Whether BEX and LIG can form a co-crystal and can improve physicochemical properties of BEX is still scant. In this study, a co-crystal of BEX and LIG, 2BEX-LIG, was generated and a rapid and sensitive LC-MS detecting method was applied to measure the concentration of BEX in plasma and tissues in order to assess whether oral administration with the novel co-crystal can enhance the bioavailability and the cerebral distribution of BEX. The new co-crystal and the new finding of its pharmacokinetic characteristics may provide a new strategy to enlarge clinical usage of BEX, especially to those of the nerve system diseases and impulse dosage forms of marketed drugs.

## 2. Materials and Methods

### 2.1. Compounds and Agents

BEX (purity > 99%, MW = 348, and pKa = 4.2 [41]) and LIG (purity > 98%, MW = 136, pKa = 3.55 [42], and sublimated at room temperature) (Figure 1) were purchased from Beijing Ouhe Techmology Co. Ltd. (Beijing, China) and Hubei Wande Chemical Co. Ltd. (Tianmen, China), respectively. Other chemicals and solvents used in this study were of analytical grade. Milli-Q Reagent water system (Walters, MA, USA) was used to purify the water provided by Hangzhou Wahaha Group Co. (Beijing, China). The solid drug delivery device (patent No: 201010219220.5) was invented and prepared by the National Drug Screening Center of the Institute of Materia Medica, Chinese Academy of Medical Sciences.

### 2.2. Experimental Animals

Sprague–Dawley (SD) rats (body weight 210 ± 10 g, Specific pathogen-free (SPF) grade) were purchased from Beijing Vital River Experimental Animal Co. Ltd. (Vital River, Beijing, China), certificate no: SCXK (Beijing) 2016-0006. These animals were kept in an environment at 24 ± 2 °C with ad libitum access to food and water. Experimental protocols were performed in accordance with institutional guidelines for the care and use of laboratory animals at the Institute of Materia Medica, Chinese Academy of Medical Science and Peking Union Medical College and at the National Institutes of Health Guide for Care and Use of Laboratory Animals (publication no. 85-23, revised 1985).

### 2.3. Construction of Ternary Phase Diagram

Ternary phase diagram of 2BEX-LIG was constructed by measuring the concentration of the saturated solutions of BEX and LIG at different ratios (Table 1). First, the density of ethanol was measured by adding 5 mL into a breaker using a pipetting gun and by weighting on a 1/1000 balance to confirm the density was suitable for ternary phase diagram. Then, the chemicals and 2 mL of ethanol were added into 5-mL sealed vials at 20 °C, and suspensions were stirred over 24 h to allow the system to reach a thermodynamic equilibrium. After filtration, solutions were diluted and analyzed by HPLC (Agilent 1200) to determine the solubility curve, and the invariant points, namely, APIs and co-crystal (C1), and LIG and co-crystal (C2) were determined by the method described by Rodriguez-Hornedo et al. [43]. Ternary phase diagrams were drawn using OriginPro 8.5 software.

### 2.4. Preparation of 2BEX-LIG and the Elimination of the Excessive LIG from the Mixture

The powder sample of 2BEX-LIG was prepared by slurry and liquid-assistant grinding, and the bulk sample used for dissolution studies and in vivo studies was prepared by liquid-assistant grinding.

Slurry: a mixture of BEX and LIG with a mole ratio of 1:5 (including BEX, 0.1 mmol, 34.8 mg and LIG, 0.5 mmol, 68.0 mg) was added to 1 mL ethanol (O1 point in Figure 2), and the suspension was stirred at a rotation speed of 300 rpm for 3 h at 20 °C. After filtration, the solid was dried at 50 °C for 30 min.

Liquid-assistant grinding: approximately 0.5 mL ethanol was added to a mixture of BEX and LIG with a mole ratio of 1:1(BEX, 0.1 mmol, 34.8 mg and LIG, 0.1 mmol, 13.6 mg), and the materials were ground at 30 Hz for 15 min at 20 °C with Retsch MM400.

Purify: the excess LIG was removed using sublimation and low melting point (about 85 °C according to the Differential Scanning Calorimeter (DSC) experiment) of LIG. Briefly, after filtration, the solid was dried at 50 °C for 30 min; in the meanwhile, the excessive LIG and the LIG attached to the surface of 2BEX-LIG were sublimated and removed during the process, so the solid obtained was pure co-crystal.

Storage and identification: subsequently, the samples were milled, sieved through 100-mesh sieves, and stored at low temperature in a dry environment until they were used for dissolution studies and in vivo studies. The powder samples were examined through Powder X-ray Diffraction (PXRD), and the patterns were compared with the calculated result of Single Crystal X-ray diffraction (SCXRD) from Mercury 2.4 to confirm the phase purity of products.

Solution crystallization: a mixture of BEX (0.1 mmol, 34.8 mg) and LIG (0.1 mmol, 13.6 mg) was added to 20 mL acetonitrile, and the suspension was stirred at a rotation speed of 300 rpm for 2 h at 20 °C. After filtration, 20 mL of cyclohexane was added to the solution, and the acetonitrile and cyclohexane were immiscible. The solution system was crystallized at 20 °C. After 30 days, a colorless plate-shaped single crystal suitable for SCXRD experiment was harvested.

### 2.5. Characterization of 2BEX-LIG

SCXRD: SCXRD was performed on a Rigaku MicroMax-002+ diffractometer (Rigaku Americas, the Woodlands, America) using monochromated Cu−Kα radiation (λ = 1.54178 Å) at 293 K. The structure was solved by OLEX2 using a direct method and refined by using a full-matrix least-squares technique. Non-hydrogen atoms were refined with anisotropic displacement parameters, and the hydrogen atoms were placed in calculated positions and refined with a riding model. Crystallographic data in the cif format were deposited in the Cambridge Crystallographic Data Center, Crystallographic Information File (CIF) and check-CIF as Appendix A were submitted, and CCDC No. 184424 is for the 2BEX-LIG co-crystal.

PXRD: PXRD experiments were performed using a Rigaku D/max-2550 diffractometer (Rigaku, Tokyo, Japan) equipped with a Cu–Kα radiation source that was set to 40 kV and 150 mA. Patterns were collected in the 2θ range of 3–40° with a scan rate of 8° min^−1^. Data were analyzed using Jade 6.0 software.

DSC: the DSC experiments were performed on a Mettler Toledo DSC/DSC1 (Mettler Toledo, Greifensee, Switzerland) using an alumina crucible. Measurement of accurately weighted samples started at 30 °C and then increased up to 300 °C with a temperature increase of 10 °C min^−1^ under a N_2_ flow of 50 mL/min.

Thermogravimetric Analysis (TGA): TGA experiment was performed on a Mettler Toledo DSC/TGA1 (Mettler Toledo, Greifensee, Switzerland) using a 40 μL alumina crucible with a flow of 50 mL/min of N_2_. Measurement of the samples started at 30 °C and then increased up to 500 °C at a temperature increase of 10 °C min^−1^. The STAR software package (STARe Default DB V9.10, Mettler Toledo, Greifensee, Switzerland) was used to record and analyze TGA and DSC profiles.

High-Performance Liquid Chromatography (HPLC): 6.96 mg BEX, 10.58 mg LIG, and 8.37 mg 2BEX-LIG were accurately weighted and dissolved in 25 mL methanol. The solutions were measured with Agilent 1200 (Agilent Technologies, Santa Clara, USA). The mobile phases were composed with methanol and water at a ratio of 85:15 (*v*:*v*), and a Kromasil 100-5-C18 column (250 mm × 4.6 mm, 5 µm) was used as the analytic column; chromatograms were monitored at 259 nm, and isocratic elution was used for all processes. The flow rate was set at 1 mL/min, the injection solvent was methanol, and the injection volume was 10 μL each time.

### 2.6. Dissolution Measurements

It was necessary that all the particle sizes of BEX and the co-crystal are at the same level when the same operation was performed on these samples. To do so, the powder samples of BEX and the co-crystal were milled and sieved through 100-mesh sieves to minimize the size influence on the results. Accurately weighed BEX (60.0 mg) and 2BEX-LIG (equivalent to 60.0 mg BEX) were added to dissolution vessels containing 900 mL of pure water or PBS (phosphate buffer saline) (pH = 6.8). Samples were stirred at 100 rpm at 37 °C, and collected at 0, 5, 15, 30, 60, 90, 120, 180, 240, 360, and 480 min. After filtration using 0.22-μm Poly tetra fluoro ethylene (PTFE) filters, the concentration was determined by HPLC (Agilent 1200). The remaining solid was examined through PXRD to study the stability during the dissolution analysis.

### 2.7. Stability Study

To investigate the stability of 2BEX-LIG, powder samples of 2BEX-LIG were placed in a high temperature (60 °C), high humidity (25 °C, 90 ± 5%), and illumination (4500 lx ± 500 lx) for 10 days. The samples were examined via PXRD analysis.

### 2.8. Pharmacokinetic Study of BEX in SD Rats

Powder samples of BEX and 2BEX-LIG prepared by liquid assistant grinding, milled, and sieved through 100-mesh sieves were used to conduct the pharmacokinetics experiments to evaluate the effects of the co-crystal. In brief, SD rats were fasted for 12 h with free access to water before the experiments. A total of 18 rats (9 males and 9 females) were randomly assigned to three groups (*n* = 6); one group of SD rats were injected through the tail vein with 5 mg/kg BEX solution, which was prepared through a two-step process. First, the BEX powder was dissolved in Dimethyl sulfoxide (DMSO) and kept at a concentration to 65 mg/mL. Second, the BEX solution dissolved in DMSO was diluted with 0.5% Sodium carboxymethylcellulose (CMC-Na) under 65–70 °C in a container to 2 mg/mL; the pH of the final form was 7.5. Another two groups of SD rats were administrated with BEX or 2BEX-LIG (equivalent to 30 mg/kg BEX), which were loaded into the solid drug delivery device and put into the stomach of rats through their mouths directly. The solid drug delivery device including a hollow needle and a rod to push drugs was invented by the National Drug Screening Center of the Institute of Materia Medica, Chinese Academy of Medical Sciences (patent No: 201010219220.5). Blood was harvested from the ophthalmic venous plexus and collected into 1.5-mL heparin-containing tubes after oral administration of BEX or 2BEX-LIG at 0, 0.25, 0.75, 1, 1.5, 2, 3, 4, 6, 7, 12, 24, 36, and 48 h and after tail vein injection of BEX at 0, 0.033, 0.083, 0.25, 0.5, 0.75, 1, 2, 3, 4, 7, 12, and 24 h. Blood samples were immediately centrifuged at 5000 rpm for 10 min at 4 °C, and supernatants were collected and stored at −40 °C for LC-MS analysis.

### 2.9. Tissue Distribution Study In Vivo

For the tissue distribution study, 84 SD rats (42 males and 42 females) were randomly divided into 14 groups (*n* = 6) and were fasted for 12 h with free access to water before the experiments. SD rats were administrated with BEX or 2BEX-LIG (equivalent to 30 mg/kg BEX), which were loaded into the solid drug delivery device and put into the stomach of rats through their mouths directly. The solid drug delivery device including a hollow needle and a rod to push drugs was invented by the National Drug Screening Center of the Institute of Materia Medica, Chinese Academy of Medical Sciences (patent No: 201010219220.5). Organs, including heart, liver, spleen, lung, kidney, and brain (cortex/other brain tissues except the cortex) were collected and immediately separated on ice at 0, 1, 2, 4, 6, 12, and 24 h after administration in the rats. In order to reduce interference of blood, the procedures that avoid contaminating tissues, especially in the rat brain, from blood during tissue sampling in rats were carefully performed. Briefly, after SD rats were sacrificed, the brain was immediately separated on ice; then, the blood was quickly rinsed with 0.9% physiological saline, and the surface water was blotted with absorbent paper; the brain tissue was then sent to a third step with the cerebral blood vessels gently stripped under a microscope; and finally the cerebral cortex and other brain tissues except the cortex were separated. The tissue samples were stored at −40 °C for LC-MS analysis.

### 2.10. Detection of BEX Concentration Using LC-MS Method

The concentration of BEX in the biological samples was detected using the LC-MS method. LC-MS/MS analytical conditions for separation were carried out using an LC-MS system equipped with Agilent 1200 liquid chromatography and 6110 mass spectrometric (Santa Clara, CA, USA). An Agilent Eclipse Plus-C18 column (2.1 × 100 mm, 3.5 µm, Agilent, Santa Clara, CA, USA) was used for separation. The mobile phases were composed of component A (acetonitrile) and component B (water (0.1% ammonium hydroxide)). Ursolic acid served as an internal standard (IS). The column temperature was maintained at 30 °C, and the injection volume was 10 µL. The flow rate of the mobile phase mixture was set at 0.3 mL/min. The 6110 Mass spectrometric was selected as negative electrospray ionization (ESI) screening with a single monitoring mode ([M-H]^−^
*m*/*z* = 347 for BEX; *m*/*z* = 455 for IS) to quantitatively analyze BEX [44]. The ion source parameters were set as follow: nebulizer pressure of 35 pounds per square inch gauge, capillary pressure of 3000 V, drying gas temperature of 350 °C, and drying gas flow rate of 10 L/min. The fragment voltages of BEX and IS were optimized to 130 V and 140 V, respectively. The detection method was validated as feasible.

Sample preparation and extraction were performed prior to LC-MS detection; biological samples were protected from light when incubated at room temperature. Aliquots of 100 μL of each plasma or tissue sample were respectively mixed with 10 μL of working IS solution (10 μg/mL). The mixture was vortexed for 2 min, and then, 300 μL acetonitrile and 500 μL ethyl acetate were successively added to extract the analytes. After vortexing for 3 min and centrifugation at 1.34 × 10^4^ rpm for 10 min, 880 μL of supernatant was transferred to a sterile clear centrifuge tube, and evaporation was performed using a slow flow of nitrogen gas. A total of 100 μL reconstitution liquid (acetonitrile: H_2_O (0.1% ammonium hydroxide) = 30:70, *v*/*v*) was added and mixed thoroughly. After centrifugation at 1.34 × 10^4^ rpm for 10 min, 10 μL of supernatant was injected into the LC-MS system for analysis.

### 2.11. Statistical Analysis

DAS 2.0 pharmacokinetic program software (Chinese Pharmacology Society, Shanghai, China) was used to analyze the pharmacokinetic parameters, including peak concentration (C_max_), peak time (T_max_), AUC, cleaning half-life (T_1/2_), and the mean residence time (MRT). All parameters were expressed as the mean ± SD. Statistical analysis and comparisons between groups were carried out using GraphPad prism software. *p* < 0.05 was considered statistically significant.

Absolute bioavailability (Fabs%) was calculated as follows:(1)$${\mathrm{Fabs}\%\;=\;100\%\;\times \;\mathrm{AUC}}_{\left(0-\infty \right)\;\mathrm{po}}{\;\times \;\mathrm{Dose}\;}_{\mathrm{iv}}{/(\mathrm{AUC}}_{\left(0-\infty \right)\;\mathrm{iv}}{\;\times \;\mathrm{Dose}}_{\;\mathrm{po}})$$
where AUC_po_ represents the area under the curve for oral administration, Dose_iv_ represents the dose of injection, AUC_iv_ represents the area under the curve for tail vein injection, and Dose_po_ represents the dose of oral administration.

## 3. Results and Discussion

### 3.1. Construction of Co-Crystal Ternary Phase Diagram

The ethanol density in our present study was 0.793 g/mL^3^, close to the standard density (0.79 g/mL^3^) at 20 °C [45], which was suitable for ternary phase diagram. The ternary phase diagram of 2BEX-LIG in ethanol was obtained at 20 °C shown in Figure 2. The phase diagram was divided into six regions, where region 1 represented the unsaturated solution phase and regions 2 (highlight in blue), 4 (highlight in red), and 6 (highlight in yellow) represented the phase equilibriums between solution and the solid phases of BEX, co-crystal, and LIG, respectively. Regions 3 and 5 represented the coexistence of co-crystal with BEX and LIG, respectively. The black lines represent the 2:1 and 1:5 molar ratios of BEX and LIG, respectively; point C represents the solid co-crystal; point O represents the mixture of BEX and LIG in molar ratio 1:5; point O1 represents the experiment condition of slurry; and C1 and C2 are invariant points in the system. The invariant points (C1 and C2) in the system were determined experimentally. Based upon the facts that the co-crystal stable region (region 4, indicating the phase equilibrium between solution and the solid phase of co-crystal) inclined significantly towards the LIG axis due to the higher solubility of LIG than BEX, the 2:1 molar ratio line came through regions 1, 2, and 3, and no intersection was shown between the 2:1 molar ratio line and region 4. By decreasing ethanol, the solid phase of BEX and LIG in co-crystal stoichiometry (2:1 ratio line) was transformed from pure BEX into the mixture of co-crystal and BEX. It was concluded that it was impossible to establish an equilibrium between the solid phase of the co-crystal and the stoichiometric solution. The results indicated that the construction of ternary phase diagram was helpful to explain the incomplete reaction of BEX with LIG in co-crystal stoichiometry and to adjust the dosage to obtain pure co-crystal in ethanol at 20 °C. Thus, excessive LIG was introduced to the reaction for the preparation of the co-crystal through slurry and liquid assistant grinding. The excessive LIG was removed at 50 °C using its sublimation and low melting point (about 85 °C according to the DSC experiment).

Phase diagram is a useful method to study the appropriate ranges of co-crystal by measuring the concentration of the saturated solubility of the compounds. For highly soluble compounds and chemicals without UV absorption [46,47], the solubility curve can be drawn by recording the amount of solvents and compounds used in preparing saturated solutions at different ratios. However, error can be brought into the results due to the volatilization of solvents and the measurement of researches on the saturated solutions. In this work, impurities and toxicity should be considered in the preparation of the pharmaceutical co-crystal. Therefore, the preparation of 2BEX-LIG was optimized in ethanol at room temperature.

### 3.2. Characterization of 2BEX-LIG

The structure of 2BEX-LIG was determined using SCXRD, a technique with high accuracy to determine chemical structures that is widely adopted and approved in previous studies [8,48,49,50,51]. As is shown in Figure 3, BEX and LIG were formed into the co-crystal mainly driven by an O-H···N hydrogen bond, where -COOH was the hydrogen donor and the N atom in pyrazine served as the hydrogen acceptor (Figure 3a). Detailed crystallographic data for the co-crystal are summarized in Table 2, indicating that a single crystal was crystallized in the C2/c space group of the monoclinic system which contained BEX and 0.5 LIG molecules in its asymmetric unit. BEX and LIG were contacted through O_2_-H_2_…N_1_, and no proton transfer occurred between -COOH and the N atom on pyrazine (Figure 3a), which furtherly confirmed the formation of 2BEX-LIG was a co-crystal instead of a salt. A layer structure is shown along the b axis (Figure 3b).

Taken together, it indicated that no hydrogen bond was formed with the carbonyl in the co-crystal compared with BEX [52], and the change of inter-molecular hydrogen bond may lead to the difference in the physicochemical properties of 2BEX-LIG.

Phase purity of the new co-crystal: PXRD, DSC, TGA, and HPLC experiments were performed to confirm the phase purity of 2BEX-LIG powder samples (Figure 4).

PXRD: the positions of characteristic peaks in PXRD patterns were furtherly detected to identify the formation of the samples. Results of the PXRD assay (Figure 4a) showed that the experimental pattern of 2BEX-LIG differed from BEX and LIG and was in good agreement with the simulated ones, thereby indicating a high phase purity of the powder samples. The molar ratios of BEX and LIG arranged as 2:1 to 1:2 were used for liquid assistant grinding with an indication that, when BEX: LIG was prepared at the molar ratio of 1:1 and 2:1 in liquid assistant grinding, the mixtures for co-crystals with LIG and BEX were obtained. The characteristic peaks of BEX in the PXRD pattern at a 2:1 molar ratio confirmed the conclusion from the ternary phase diagram that BEX and LIG did not react completely in co-crystal stoichiometry. It was in accordance with the results obtained from the analysis of the ternary phase diagram that the products of the ratio in region 4 in Figure 2 prepared through slurry were high-phase purity co-crystals.

DSC and TGA: In the DSC patterns, as is shown in Figure 4b, the endothermic peak of BEX was at 224.33 °C, and those of LIG were at 87.74 and 185.72 °C. The endothermic peaks of 2BEX-LIG appeared at 133.45 and 223.49 °C. The TGA of 2BEX-LIG was carried out as shown in Figure 4d. The weight loss in the profile was 15.46%, equivalent to 0.95 LIG molecules lost in the co-crystal formation.

HPLC: The retention times of LIG and BEX were kept at 3 and 22 min, respectively. The peak area of the samples is summarized in Table 3, and the concentrations of BEX and LIG were 8.017 × 10^−4^ and 4.072 × 10^−4^ mmol/mL, respectively, calculated from the data, which indicated the molar ratio in the co-crystal was BEX–LIG = 1.97:1.

To confirm the transformation of 2BEX-LIG at the first endothermic peak, the co-crystal was heated to 150 °C and examined through PXRD, confirming that the co-crystal was transformed into BEX, which indicated that LIG was lost before the melting point (223.49 °C) of BEX because of its sublimation and low melting point and that the co-crystal was decomposed during the process (Figure 5).

### 3.3. Powder Dissolution Measurements

BEX is an API of poor water solubility. Improved solubility can lead to increased bioavailability [12,53]. The dissolution experiment was performed based on General rule 0931 of the fourth part of Chinese Pharmacopoeia [54] and published literature [6,11,55]. The dissolutions of BEX and 2BEX-LIG were about 0.24 and 2.90 μg/mL (0.36% and 4.36% of the sample added to media) in pure water, respectively, as shown in Figure 6a. The maximum concentration of 2BEX-LIG was approximately 12.1-fold higher than BEX, and the improved dissolution last for over 8 h, indicating enhanced bioavailability. However, the dissolution of 2BEX-LIG was not improved significantly compared with BEX in PBS such that the maximum concentrations of BEX and 2BEX-LIG were 0.19 and 0.20 μg/mL (0.28% and 0.29% of the sample added to media) (Figure 6b), respectively. The remaining solid was examined through PXRD to study the stability during the dissolution analysis (Figure 7). The co-crystal was stable in pure water, and part of 2BEX-LIG was transformed into BEX in PBS during the process, which could explain the failure to improve the solubility in PBS. BEX may decrease the contact area of the media, and the co-crystal affects the solubility of 2BEX-LIG. The solubility could be affected by the layer structure through changing inter-molecular hydrogen bonds and display of the chemicals [8,56,57,58,59].

### 3.4. Stability Study

The stability of 2BEX-LIG at high temperature (60 °C), high humidity, and illumination for 10 days was investigated according to General rule 9001 of the fourth part of Chinese Pharmacopoeia, which was widely used in previous studies [60] in order to estimate the primary stability of the co-crystal. The samples were then examined through PXRD, as presented in Figure 4c. 2BEX-LIG was transformed into BEX due to the sublimation of LIG at high temperature and was stable both in high humidity and illumination conditions. Hence, it indicates that 2BEX-LIG is suitable for hypothermic preservation owing to its instability at high temperature resulting from the sublimation of LIG. It is in accordance with a previous study, which showed that a co-crystal of ethinyl estradiol and LIG was transformed into amorphous at 100 °C and changed to ethinyl estradiol at 130 °C [61].

### 3.5. Pharmacokinetic Study

The pharmacokinetic study was performed and analyzed by DAS 2.0 (non-compartment model) software. The mean plasma concentration–time curves in rats that received intravenous (i.v.) BEX (5 mg/kg) and oral BEX (30 mg/kg) or 2BEX-LIG (equivalent to 30 mg/kg BEX) are presented in Figure 8, and the pharmacokinetic parameters are listed in Table 4. C_max_ of BEX in the plasma sample of rats administered 2BEX-LIG was roughly 2-fold higher than those of rats administrated with BEX, and T_max_ was delayed from 7.33 h to 9.58 h, while MRT did not significantly change from 11 to 11.33 h. AUC_(0-∞)_ of BEX in rats administrated 2BEX-LIG was significantly improved from 7.30 × 10^3^ to 1.31 × 10^4^ μg/L·h, and absolute bioavailability of BEX about 2-fold in rats, which was similar to the result of the BEX nanocrystal enhanced the bioavailability of BEX, but it did not need to select optimal stabilizers, which were essential to prepare the BEX nanocrystal [25,31]. Besides, 2BEX-LIG enhancing the absolute bioavailability of BEX could be explained by the different characteristics of intermolecular arrangement order [62] between BEX and 2BEX-LIG which further improved the dissolution and absorption in intestine. However, the mechanism of 2BEX-LIG enhancing the absorption in intestine needs further studies.

### 3.6. Tissue Distribution

The concentration of BEX in various tissues is presented in Figure 9. After oral administration of BEX or the 2BEX-LIG co-crystal, T_max_ of BEX in different tissues was 4–7 h and the concentration of BEX in tissues of rats administrated 2BEX-LIG was higher than those of BEX in tissues of rats administrated BEX, especially in the liver, kidney, and brain. The co-crystal improved the concentration of BEX by 2.27-fold in the liver and by 1.88-fold in the kidney. Interestingly, C_max_ in the cortex and other brain tissues except the cortex increased from 476.34 to 1655.86 ng/g and 714.39 to 2437.42 ng/g, respectively (Figure 10). As we have pointed out, the co-crystal can impact various aspects of drug pharmacokinetics, including but not limited to drug absorption. The diversity of solid forms offered through co-crystallization can facilitate drastic changes in solubility and pharmacokinetics. Therefore, it is unsurprising that the co-crystal led to a higher bioavailability with higher concentrations in the liver and kidney [63]. However, when the concentrations in the liver and kidney are elevated to higher levels, the concentrations in the target organs can be increased to even much higher levels [64]. We have calculated tissue distribution parameter Kp of BEX using C_tissues_/C_plasma_ ratios. As is shown in Table 5, compared with BEX, 2BEX-LIG significantly improved Kp by about 2-fold in the cortex and in the brain, significantly higher than in the liver and kidney, indicating that a relatively low dose will reach effective concentrations in the brain and cortex without increasing the toxicity. It is obviously that 2BEX-LIG may benefit brain-related diseases.

In previous studies, the tissue concentration of BEX in rats administrated BEX nanocrystals and the treatment of lung cancer have been reported [30], but the concentration of BEX in the brain and the potential treatment of brain cancer was not discussed. In this study, we present 2BEX-LIG as the first co-crystal of BEX with high potential to treat brain cancer, which may help to provide novel conceptual insight into the field of anti-brain disease drug R&D. Besides, the concentration of BEX in the cerebrum of rats administrated 2BEX-LIG offered the foundation of pharmacodynamics for treatment of nervous system diseases, and the concentration lasted for 24 h, indicating that it has the potential to maintain a longer effect, which provides preclinical data support for the development of new drugs to treat brain diseases.

## 4. Conclusions

In this study, there were two major findings. First, a drug–drug co-crystal, of which BEX is the API, was designed with a superior dissolution and a satisfactory stability. The ternary phase diagram was constructed to study the reasonable method to prepare a high phase purity co-crystal. Second, the new co-crystal of 2BEX-LIG as a new chemical entity expressed a much higher absolute bioavailability and an improved tissues distribution especially with a significant increase in intra-cerebral distribution, thereby providing potential for clinical therapy of brain tumors and other diseases. Hence, our present study indicates that the novel co-crystal contributes to improving pharmacokinetic characteristics of BEX, which provides a new strategy to use dosage forms of marketed drugs.

Besides, it is possible that enhanced levels of systemic and cerebral exposure to BEX from increased doses of the co-crystal can result from any drug metabolizing enzyme and/or drug transporter-mediated LIG–BEX interaction, and it as a very interesting hypothesis that need to be verified in the future research.

## Figures and Tables

**Figure 1 pharmaceutics-12-00906-f001:**
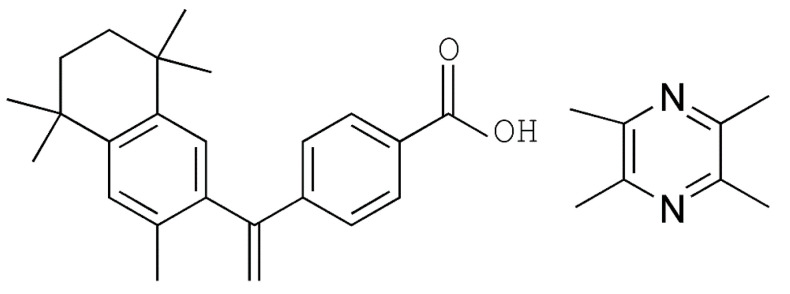
Chemical structure of bexarotene (BEX) (**left**) and ligustrazine (LIG) (**right**).

**Figure 2 pharmaceutics-12-00906-f002:**
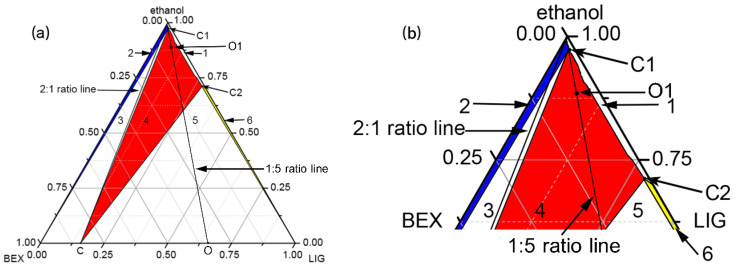
Ternary phase diagram of 2bexarotene- ligustrazine (2BEX-LIG) in ethanol at 20 °C drawn with OriginPro 8.5 software: the phase diagram was divided into six regions, where region 1 represents the unsaturated solution phase and regions 2 (highlight in blue), 4 (highlight in red), and 6 (highlight in yellow) represents the phase equilibrium between solution and the solid phases of BEX, co-crystal, and LIG, respectively. Regions 3 and 5 represent the coexistence of the co-crystal with BEX and LIG, respectively. The black lines represent the 2:1 and 1:5 molar ratios of BEX and LIG, respectively; point C represents the solid co-crystal; point O represents the mixture of BEX and LIG in molar ratio 1:5; point O1 represents the experiment condition of slurry; and C1 and C2 are invariant points in the system. (**a**) The full image; (**b**) zoom on the upper part (about 1.7-fold magnified).

**Figure 3 pharmaceutics-12-00906-f003:**
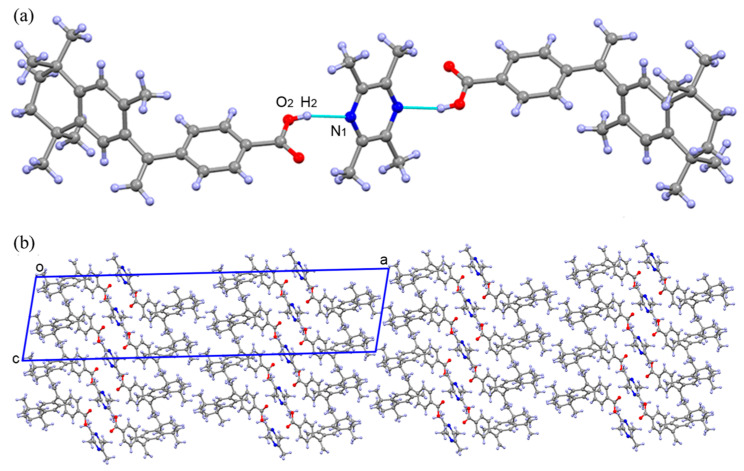
Single crystal X-ray diffraction of 2bexarotene-ligustrazinewas (2BEX-LIG) drawn with Mercury 2.4, where the red balls are O atoms, blue balls are N atoms, gray balls are C atoms, and white balls are H atoms: the blue squares were the asymmetric unit of 2BEX-LIG along the a, b, and c axes. (**a**) Hydrogen bond between O_2_-H_2_···N_1_; (**b**) a layer structure along the b axis.

**Figure 4 pharmaceutics-12-00906-f004:**
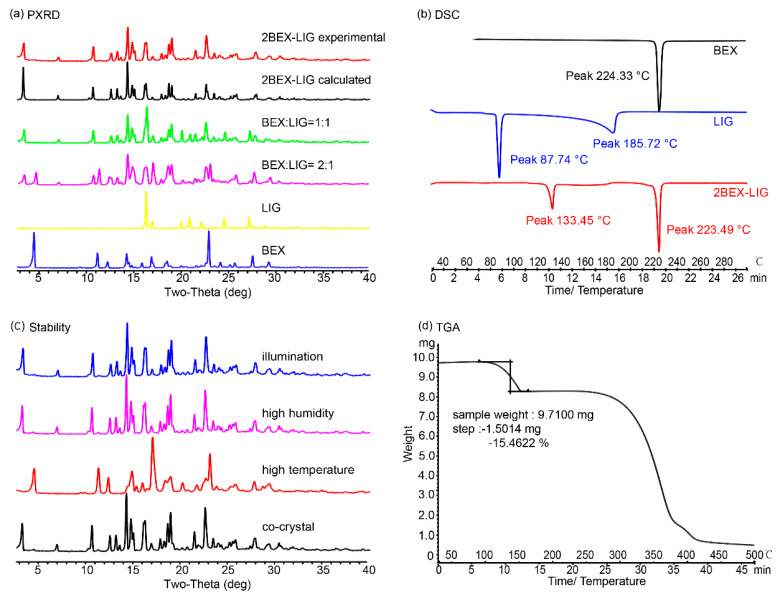
Characterization of 2bexarotene-ligustrazine (2BEX-LIG): (**a**) Powder X-ray diffraction (PXRD) patterns of experimental and calculated 2BEX-LIG, products of original chemicals in 1:1 and 2:1, BEX, and LIG; (**b**) Differential Scanning Calorimeter (DSC) patterns of BEX (black), LIG (blue), and 2BEX-LIG (red); (**c**) PXRD patterns of 2BEX-LIG in high humidity, illumination, high temperature, and pure co-crystal performed with the same method as in (**a**); and (**d**) Thermogravimetric Analysis (TGA) pattern of 2BEX-LIG.

**Figure 5 pharmaceutics-12-00906-f005:**
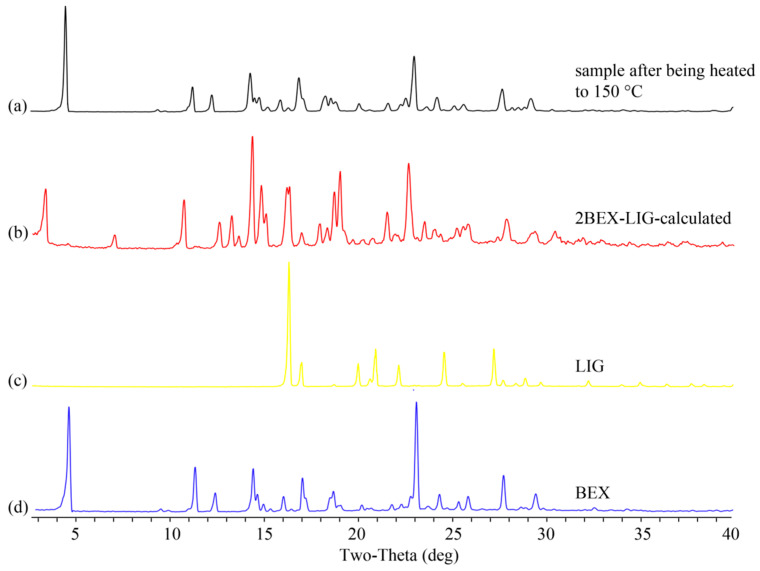
Powder X-ray diffraction (PXRD) patterns of the samples: (**a**) sample after being heated to 150 °C; (**b**) calculated 2bexarotene-ligustrazine (2BEX-LIG); (**c**) ligustrazine (LIG); and (**d**) bexarotene (BEX).

**Figure 6 pharmaceutics-12-00906-f006:**
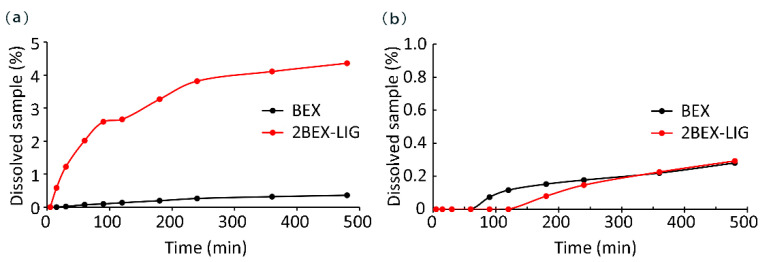
Powder dissolution profiles of bexarotene (BEX) and 2bexarotene-ligustrazine (2BEX-LIG) (**a**) in pure water and (**b**) in phosphate buffer saline (PBS).

**Figure 7 pharmaceutics-12-00906-f007:**
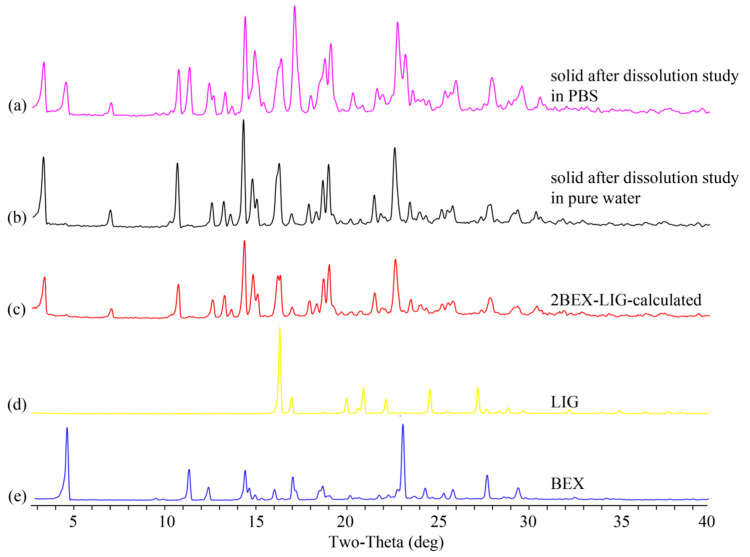
Powder X-ray diffraction (PXRD) patterns of the samples: (**a**) sample after dissolution study in phosphate buffer saline (PBS); (**b**) solid after dissolution study in pure water; (**c**) calculated 2bexarotene-ligustrazine (2BEX-LIG); (**d**) ligustrazine (LIG); and (**e**) bexarotene (BEX).

**Figure 8 pharmaceutics-12-00906-f008:**
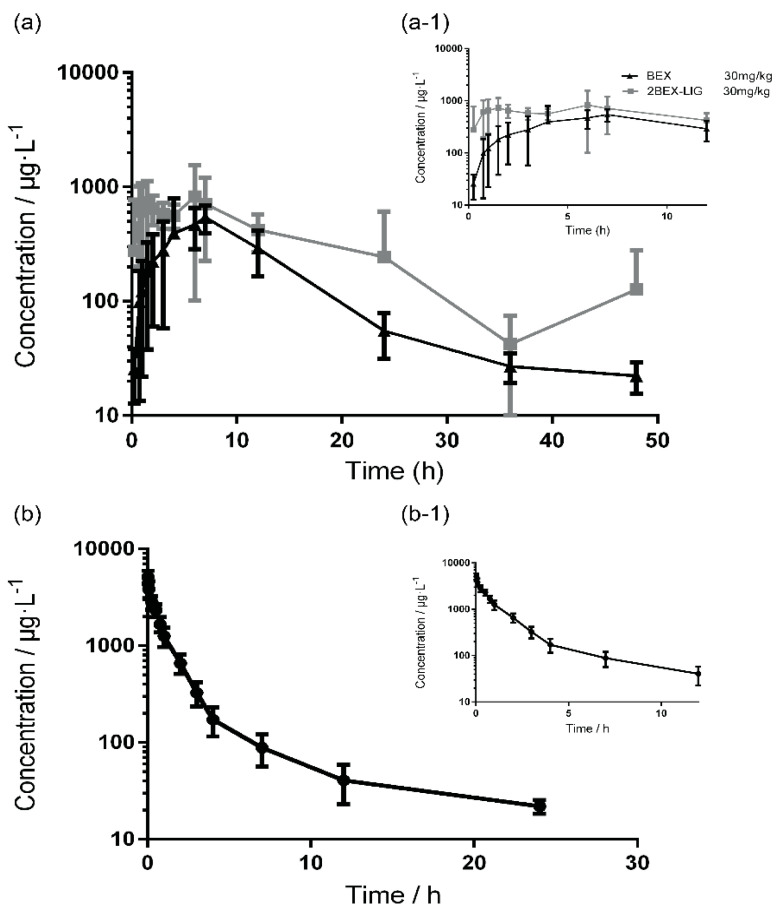
The mean plasma concentration–time curve of bexarotene (BEX) in SD rats: (**a**) orally administrated with BEX (30 mg/kg) and 2bexarotene-ligustrazine (2BEX-LIG) (equivalent to 30 mg/kg BEX); (**a-1**) the magnification of the concentration-time curve before 12 h; (**b**) the mean plasma concentration–time curve of SD rats after tail vein injection of BEX (5 mg/kg); and (**b-1**) the magnification of the concentration–time curve before 12 h. *n* = 6.

**Figure 9 pharmaceutics-12-00906-f009:**
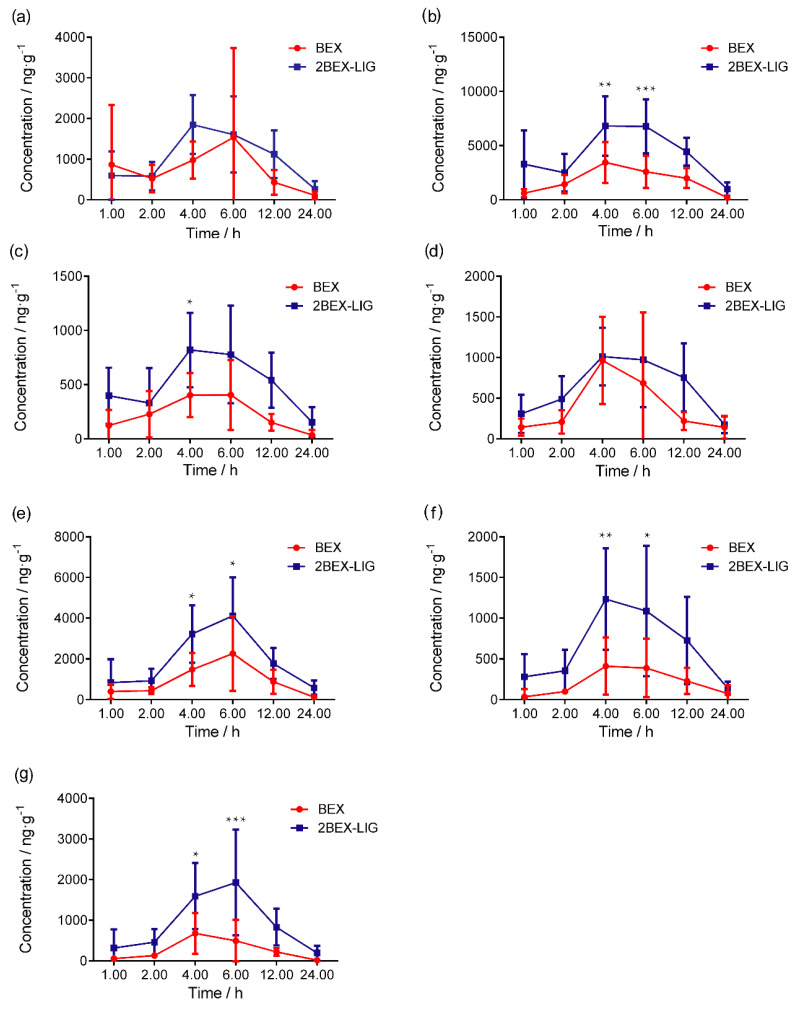
The concentration of bexarotene (BEX) in tissues of SD rats orally administrated BEX (30 mg/kg) and 2bexarotene-ligustrazine (2BEX-LIG) (equivalent to 30 mg/kg BEX) at 1, 2, 4, 6, 12, and 24 h in (**a**) the heart; (**b**) the liver; (**c**) the spleen; (**d**) the lungs; (**e**) the kidneys; (**f**) the cortex; and (**g**) others _(brain)_. (*n* = 6). * *p* < 0.05, ** *p* < 0.01, and *** *p* < 0.001 vs. BEX.

**Figure 10 pharmaceutics-12-00906-f010:**
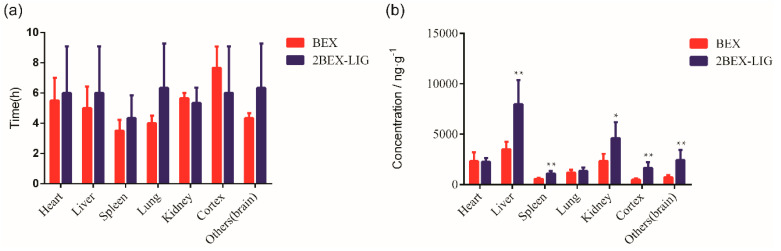
Tissues distribution parameters of bexarotene (BEX) in SD rats orally administrated with BEX (30 mg/kg) and 2bexarotene-ligustrazine (2BEX-LIG) (equivalent to 30 mg/kg BEX): (**a**) the peak time of maximum concentration in tissues and (**b**) the maximum concentration of BEX in tissues. (*n* = 6). * *p* < 0.05, ** *p* < 0.01 vs. BEX.

**Table 1 pharmaceutics-12-00906-t001:** Initial experimental data for system of 2bexarotene-ligustrazine (2BEX-LIG) ethanol at 20 °C.

Weight (mg)		Concentration (mg/mL)		Weight (mg)		Concentration (mg/mL)	
LIG	BEX	LIG	BEX	LIG	BEX	LIG	BEX
0.00	40.00	0.00	7.01	120.00	40.00	39.09	9.44
3.00	40.00	0.50	6.57	140.00	40.00	46.32	11.74
6.00	40.00	2.49	6.54	160.00	40.00	47.51	12.78
9.00	40.00	4.00	7.84	180.00	40.00	47.89	12.03
12.00	40.00	5.38	7.57	200.00	40.00	48.11	12.12
15.00	40.00	5.87	6.77	220.00	40.00	48.55	12.71
18.00	40.00	6.91	7.50	240.00	40.00	51.16	11.76
21.00	40.00	7.07	7.09	260.00	40.00	51.86	13.06
24.00	40.00	8.44	7.36	280.00	40.00	55.50	14.42
27.00	40.00	9.46	6.81	300.00	40.00	56.14	13.55
30.00	40.00	9.58	7.47	320.00	40.00	57.04	13.77
40.00	40.00	11.85	8.90	340.00	40.00	58.51	14.31
50.00	40.00	14.75	8.19	360.00	40.00	61.60	14.87
60.00	40.00	17.84	9.81	380.00	40.00	66.26	15.99
70.00	40.00	19.88	9.45	400.00	40.00	74.23	14.25
80.00	40.00	22.98	9.40	600.00	40.00	236.13	16.47
90.00	40.00	30.46	10.68	800.00	40.00	249.90	13.29
100.00	40.00	30.58	10.13	1400.00	40.00	312.43	10.96
110.00	40.00	34.54	10.07	3000.00	0.00	332.64	0.00

BEX: bexarotene; LIG: ligustrazine.

**Table 2 pharmaceutics-12-00906-t002:** Crystallographic data of the 2bexarotene-ligustrazine (2BEX-LIG) co-crystal.

Crystallographic Data	2BEX-LIG
Empirical formula	C_28_H_34_NO_2_
Temperature (K)	293
Crystal system	monoclinic
Space group	C2/c
a (Å)	49.281
b (Å)	8.517
c (Å)	11.787
α (deg)	90.00
β (deg)	100.41
γ (deg)	90.00
Volume (Å^3^)	4866
Z	8
Calculated density (g/cm^3^)	1.137
Absorption coefficient. (mm^−1^)	0.546
F (000)	1800
Crystal size (mm)	0.2 × 0.2 × 0.2
Rint	0.0710
R1 [I>2σ(I)]	0.0621
wR2 (all data)	0.1693
GOF	0.976

GOF: goodness of fit

**Table 3 pharmaceutics-12-00906-t003:** Data of HPLC experiments.

Samples	Retention Time (min)	Peak Area	Concentration (mg/mL)	Concentration (mmol/mL)
BEX	22.52	3.46 × 10^3^	0.28	7.99 × 10^−4^
LIG	3.07	1.16 × 10^3^	0.42	3.11 × 10^−3^
BEX in co-crystal	22.22	3.47 × 10^3^	0.28	8.02 × 10^−4^
LIG in co-crystal	3.07	1.52 × 10^2^	0.06	4.07 × 10^−4^

HPLC: High Performance Liquid Chromatography; BEX: bexarotene; LIG: ligustrazine

**Table 4 pharmaceutics-12-00906-t004:** Pharmacokinetic parameters of bexarotene (BEX) in Sprague–Dawley (SD) rats.

Parameter	Unit	Oral Dose (30 mg/kg)		Injection Dose (5 mg/kg)
		BEX	2BEX-LIG	BEX
AUC_(0–t)_	(×10^3^) μg/L h	7.03 ± 2.27	13.05 ± 3.85 **	5.14 ± 0.96
AUC_(0–∞)_	(×10^3^) μg/L·h	7.24 ± 2.27	13.57 ± 3.98	5.27 ± 0.97
T_1/2_	h	7.81 ± 2.21	7.92 ± 3.18	4.45 ± 1.53
T_max_	h	7.33 ± 2.58	9.58 ± 7.81	0.033
C_max_	(×10^3^) μg/L	0.63 ± 0.30	1.24 ± 0.60 *	5.14 ± 0.79
MRT_(0–t)_	h	11.33 ±1.30	11.00 ±4.03	2.50 ± 0.69
F_AUC(0–∞)_ %		22.89	42.86 **	/

AUC: the area under the curve; T_1/2_: cleaning half-life; T_max_: peak time; C_max_: peak concentration; MRT_(0-t)_: the mean residence time; and F_AUC(0-∞)_: absolute bioavailability; * *p* < 0.05, ** *p* < 0.01 vs. BEX. Data are means ± SD.

**Table 5 pharmaceutics-12-00906-t005:** The tissue distribution parameter Kp (C_tissues_/C_plasma_ ratios) of bexarotene (BEX) in rats.

Kp (C_tissues_/C_plasma_)	BEX	2BEX-LIG
Kp (Heart)	3.70	1.82
Kp (Liver)	5.58	6.42
Kp (Spleen)	0.91	0.88
Kp (Lung)	1.88	1.08
Kp (Kidney)	3.66	3.71
Kp (Cortex)	0.76	1.33
Kp (Others (Brain))	1.14	1.96

BEX: bexarotene; 2BEX-LIG: 2bexarotene-ligustrazine.

## Data Availability

All experimental data required to reproduce the findings from this study will be made available to interested investigators.

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
