# Peer review of "A Novel Co-Crystal of Bexarotene and Ligustrazine Improves Pharmacokinetics and Tissue Distribution of Bexarotene in SD Rats"

_pharmaceutics, 2020, doi:10.3390/pharmaceutics12100906_

Round 1

Reviewer 1 Report

Thanks for the revision and I am happy with the modified manuscript and the added experimental evidence.

Author Response

Thanks for the revision and I am happy with the modified manuscript and the added experimental evidence.

[Response] Thanks for your warm work.

Reviewer 2 Report

All the revisions are satisfactory and the manuscript is now suitable for publication.

Author Response

All the revisions are satisfactory and the manuscript is now suitable for publication.

[Response] Thanks for your warm work.

Reviewer 3 Report

General Comments to the Authors on the revised Manuscript: The authors conduct very good research and present their results well. The revised manuscript is well written but still needs a professional English language edit. The authors addressed this reviewer’s comments and added some experimental analytical results by HPLC to verify the stoichiometry. The authors put in much effort in this revised manuscript and should be congratulated.

Specific Comments to the Authors on the revised Manuscript:

  1. The authors added the weak acid pKa value of 4.2 for bexarotene to line 85, but not for ligustrazine (tetramethytpyrazine). The weak base pKa of pyrazine is ~ 0.5. Thus a salt cannot form since at no pH value the anion of bexarotene and the cation of ligustrazine exits in the same solution. Please state that a salt cannot be formed between bexarotene and ligustrazine.
  2. HPLC conditions lines 165-176, add the flow rate, injection solvent, and injection volume.
  3. Significant figures in the tables 3 and 4: Up to seven significant figures in Figure 4. Seems too many, especially for measured PK values. How many significant figures do your experimentation measurements allow?
  4. Awkward added paragraphs lines 115-120, 337-340.

Reviewer 4 Report

As the authors have addressed all of my comments properly, the manuscript can be accepted with minor English editing that can be done by journal editor during proof reading stage.

The authors have significantly improved the manuscript and the presentations, so the manuscript is suitable for your reputable journal.

I hope in the next paper, author can provide full cif that can be easily prepared by the latest OLEX2 or SHELXLE that is freely available.

Author Response

This manuscript is a resubmission of an earlier submission. The following is a list of the peer review reports and author responses from that submission.

Round 1

Reviewer 1 Report

Manuscript titled 'A novel co-crystal of bexarotene and ligustrazine improves pharmacokinetics and tissue distribution of bexarotene in SD rats' revealed the cocrystal structure and the physicochemical properties along with the pharmacokinetic studies. 

The bexarotene is a crucial compound with low solubility and it is a very good opportunity to improve this Class 4 compound. All the analytical results were clearly presented and the results look very promising.

Please consider following comments before addressing:

  1. Have authors tried any other compounds for the screening, if yes that needs to be discussed. Also, is the Ligustrazine selected randomly, what is the rationale behind this?
  2. Have authors did the stability study during the solubility and dissolution analysis? it is very important to find the stability of the cocrystal after the study.
  3. Figure 4d should be TGA not DGA
  4. The DSC thermogram has two endothermic peaks have you checked the polymorphic possibility? it can be done as an easy experiment by heating the compound and recording PXRD.

Reviewer 2 Report

In this manuscript, the authors have reported a multi-drug cocrystal consisting of bexarotene and ligustrazine. The cocrystal was thoroughly characterized and its properties, such as dissolution, stability, tissue distribution, and oral bioavailability were measured. Ternary phase diagram was constructed that provide operating conditions for solution crystallization to achieve the pure cocrystal. The cocrystal was found to show superior properties than bexarotene. The manuscript was well-written and it is suitable for publication in the journal ‘Pharmaceutics’. However, I recommend the authors to revise the manuscript based on the minor comments provided.

  1. The relevance of combining bexarotene and ligustrazine as a multi-drug cocrystal should be highlighted. Are there potential synergistic pharmacological benefits of using this combination?

  1. Did the authors characterize the samples remained after the dissolution experiments? Provide PXRD analysis of the remained samples which would provide valuable information on the stability of the cocrystal in dissolution media.

Reviewer 3 Report

General Comments to the Authors: The authors explore cocrystals of bexarotene (BEX) with the conformer ligustrazine (LIG), identify a cocrystal (2BEX-LIG) that is further investigated by solid-state characterization, dissolution, chemical stability, and in-vivo oral pharmacokinetics in rats. The authors report the in-vivo benefits of the identified cocrystal (2BEX-LIG) after oral administration in rats to include an increase in oral bioavailability from 23% to 43% and an increase in brain concentration by 3.4-fold. This reviewer’s main scientific critic is that one of the author’s main claims is that the cocrystal improves the in-vivo tissue “distribution” of bexarotene, but perhaps the better scientific claim is that the cocrystal improves the in-vivo tissue “concentration”. An intact cocrystal can improve the in-vivo absorption, but after absorption the cocrystal will dissociate to bexarotene and ligustrazine thus bexarotene will distribute into tissues to the same extent as non-cocrystal bexarotene. However, since a cocrystal can improve the in-vivo absorption the in-vivo plasma concentration can be higher than after oral administration of a non-cocrystal bexarotene, and thus a higher concentration in tissues.

Specific Comments to the Authors:

  1. To show that a salt cannot be formed between bexarotene and ligustrazine it would be useful to include the weak acid pKa value of bexarotene, and the weak base pKa value of ligustrazine.
  2. Specify if it is a mole ratio or mass ratio in the cocrystal 2BEX-LIG.
  3. The author should verify the 2:1 mole ratio by doing HPLC analysis. The PXRD data is not compelling to define the 2:1 mole ratio.
  4. It would be useful to report the sublimation temperature of ligustrazine since the authors rely on sublimation of ligustrazine for purification and is also utilized in converting 2BEX-LIG into BEX.
  5. Significant figures in the tables: Up to 6 significant figures. Seems too many, especially for measured PK values. How many significant figures do your experimentation measurements allow?
  6. Experimental conditions for intravenous: Describe the intravenous administration including formulation, pH, and dose volume.
  7. Specify the Figure 6b is after intravenous administration.
  8. Experimental condition: What is the final form of BEX that was administered orally via the solid drug delivery device? Meaning the free acid or a salt form (i.e., sodium salt)? How would these results be different if a salt from of BEX was used?
  9. Experimental conditions for dissolution (see next comment): The dissolution was conducted in water (no pH adjustment). To create sink conditions for the anionic bexarotene the dissolution media should be buffered to ~ pH 7, which is above its weak acid pKa
  10. Figure 5 dissolution: The y-axis should be % dissolved. It seems that the % dissolved is quite low (~ 4-5%), for the plot plateaus at ~ 3 µg/mL but the amount added was 60 mg bexarotene and 60 mg bexarotene equivalents of cocrystal (2BEX-LIG) in900 mL water, thus the total concentration is ~ 66 µg/mL.  You may want to repeat the dissolution under sink conditions using a pH 7 buffer where bexarotene is more soluble than in non pH-adjusted water.
  11. The PK plots: The y-axis is usually a Log10 scale.
  12. Figure 7h should be a separate plot and is a visual of the Cmax values in the other plots in Figure 7 and the Table 4 Cmax values.
